# Seroprevalence of SARS-CoV-2 specific IgG antibodies in District Srinagar, northern India – A cross-sectional study

S. Muhammad Salim Khan[1], Mariya Amin Qurieshi[1], Inaamul Haq[1]*, Sabhiya Majid[2], Arif Akbar Bhat[2], Sahila Nabi[1], Nisar Ahmad Ganai[1], Nazia Zahoor[1], Auqfeen Nisar[1], Iqra Nisar Chowdri[1], Tanzeela Bashir Qazi[1], Rafiya Kousar[1], Abdul Aziz Lone[1], Iram Sabah[1], Shahroz Nabi[1], Ishtiyaq Ahmad Sumji[1], Misbah Ferooz Kawoosa[1], Shifana Ayoub[1]

1 Department of Community Medicine, Government Medical College Srinagar, Srinagar, India, 2 Department of Biochemistry, Government Medical College Srinagar, Srinagar, India

* haqinaam@yahoo.co.in

**Data Availability Statement:** All relevant data are within the manuscript and its Supporting Information files.

## Abstract

### Background

Prevalence of IgG antibodies against SARS-CoV-2 infection provides essential information for deciding disease prevention and mitigation measures. We estimate the seroprevalence of SARS-CoV-2 specific IgG antibodies in District Srinagar.

### Methods

2906 persons >18 years of age selected from hospital visitors across District Srinagar participated in the study. We tested samples for the presence of SARS-CoV-2 specific IgG antibodies using a chemiluminescent microparticle immunoassay-based serologic test.

### Results

Age- and gender-standardized seroprevalence was 3.6% (95% CI 2.9% to 4.3%). Age 30–69 years, a recent history of symptoms of an influenza-like-illness, and a history of being placed under quarantine were significantly related to higher odds of the presence of SARS-CoV-2 specific IgG antibodies. The estimated number of SARS-CoV-2 infections during the two weeks preceding the study, adjusted for test performance, was 32602 with an estimated (median) infection-to-known-case ratio of 46 (95% CI 36 to 57).

### Conclusions

The seroprevalence of SARS-CoV-2 specific IgG antibodies is low in the District. A large proportion of the population is still susceptible to the infection. A sizeable number of infections remain undetected, and a substantial proportion of people with symptoms compatible with COVID-19 are not tested.

**Funding:** This work was supported by primarily Institutional funding with support from the District Disaster Management Authority, Srinagar.

**Competing interests:** The authors have declared that no competing interests exist.

## Introduction

Coronavirus disease (COVID-19) is an infectious disease caused by the most recently discovered coronavirus, Severe Acute Respiratory Syndrome Coronavirus 2 (SARS-CoV-2) [1]. The disease was declared as a Public Health Emergency of International Concern and later as a global pandemic by the World Health Organization [2, 3]. COVID-19 presents with flu-like symptoms and causes severe symptoms in some cases. As of 31st July 2020, there were over 17 million reported cases of SARS-CoV-2 infection, with nearly 669 thousand deaths globally [4]. In India, as of 31st July 2020, the total number of reported SARS-CoV-2 infections was over 1.6 million, with more than 35 thousand deaths [4]. In District Srinagar, the first case of SARS-CoV-2 infection was reported on 18th March 2020. On 22nd March 2020, the Jammu and Kashmir government ordered the shutdown of all non-essential activities, commercial and business establishments, and educational institutions, except essential commodities and services to prevent the spread of SARS-CoV-2 infection [5]. The health authorities started extensive case-detection and contact-tracing activities. Case-detection was based on the testing of nasopharyngeal samples by reverse transcriptase-polymerase chain reaction (RT-PCR).

People with SARS-CoV-2 infection may remain asymptomatic or develop mild to severe COVID-19, which may result in death. A large proportion of SARS-CoV-2 infections remain asymptomatic and contribute to disease transmission [6, 7]. Testing strategy and the number of test kits available influences the detection of cases by RT-PCR. RT-PCR-based case detection strategies may not provide a reasonable approximation of the number of SARS-CoV-2 infections in a population since they may miss many asymptomatic and pre-symptomatic infections. Thus, an informed policy- and decision-making for control of the COVID-19 epidemic in a community should not be based solely on RT-PCR-based numbers.

Seroprevalence surveys can provide an estimate of the proportion of the population that has developed antibodies against SARS-CoV-2, an indication of recent SARS-CoV-2 infection. Mild and asymptomatic infections, which may not have received RT-PCR testing, can be detected. Besides, assuming that antibodies provide partial or total immunity, seroprevalence surveys give an estimation of the proportion of the population still susceptible to the infection. Seroprevalence studies provide important complementary information to frame evidence-based strategies for SARS-CoV-2 infection prevention.

Here, we present the results of a cross-sectional seroprevalence study in District Srinagar, conducted between 1st and 15th July 2020, to estimate the prevalence of IgG antibodies against SARS-CoV-2 among adults using a sensitive and specific chemiluminescent microparticle immunoassay (CMIA)-based test.

## Materials and methods

### Study design, setting, and participants

We conducted a cross-sectional seroprevalence study in District Srinagar over two weeks from 1st July 2020 to 15th July 2020. District Srinagar is a city in the valley of Kashmir in northern India. It has an estimated adult (>18 years) population of just over one million. Study participants were adults (>18 years) who visited select hospitals across the District during the study period.

### Ethics statement

We informed the participants of the study's purpose and procedure. We obtained written informed consent from those who agreed to participate. The Institutional Ethics Committee of Government Medical College Srinagar approved the study.

## Sample size

We estimated the minimum sample size needed based on an anticipated seroprevalence of 2% within an absolute error of 0.8% with 95% confidence. We used a design effect of 2 to adjust for the nature of sampling and increased the sample size further to account for a non-response of 20%. The minimum sample size needed for the study was 2821. We targeted a sample size of 3000.

## Selection of participants

There are 130 hospitals across District Srinagar which provide primary, secondary, and tertiary care to the populace. We purposively selected 20 hospitals across the District so that the chosen hospitals spread across all areas of the District. Furthermore, hospitals with very meager patient visits (specifically, we excluded hospitals with less than 100 patient visits per month) were not selected. Details about the hospitals across the District are in the supplementary material (S1 Data).

We deemed hospital-based selection of participants to be more convenient, rapid, and feasible, given the constrained human resources and time available for completion of the study. Such a choice could, however, lead to a non-representative sample. We made efforts to reduce this bias by reporting age- and gender-standardized prevalence.

We invited all adult patients (>18 years) coming to the selected hospitals during the study period for participation in the study.

## Procedure

Consenting adults were required to answer a set of questions which included information about demographic variables, self-reported history of contact with a known SARS-CoV-2 positive patient, self-reported history of travel outside the valley since 1st January 2020, history of symptoms suggestive of an Influenza-Like Illness (ILI) (Fever and Cough) during the two weeks preceding the interview, history of an RT-PCR for SARS-CoV-2 infection and the result of such a test, if done. Each participant was assigned a five-digit unique identification number. Participants were interviewed by doctors who had training and experience in conducting survey interviews and were specifically trained to use the EpiCollect5 platform for recording and uploading the responses. EpiCollect5 is a free, mobile application-based tool which enables the interviewers to record and store the participant responses offline and upload them to the EpiCollect5 server from where stored data is downloaded for data analysis [8].

Upon completion of the interview, trained laboratory technicians (phlebotomists) collected 3–5 mL of venous blood from each participant under standard aseptic precautions. The blood samples were immediately transferred into a red-top serum tube with a clot activator. The samples were allowed to stand for at least 30 minutes for clotting to take place. Afterward, the samples were centrifuged at the same hospital at 3000 RPM for 10 minutes. In case the centrifugation facility was not available at the hospital, centrifugation was done at a central facility. Centrifuged blood samples were transported under the cold chain (vaccine carrier with ice packs) to a central laboratory for further processing and testing. The samples were carefully packed in designated vaccine carriers to avoid hemolysis during transportation.

## Laboratory procedure

We performed the test using Fully Automated High Throughput Platform ARCHITECT i1000SR Immunoassay Analyzer by Abbott Laboratories Inc [9]. Before testing, calibration was performed to create assay control. Once calibration was accepted and stored, serum

samples were tested for SARS-CoV-2 specific IgG. For quality control, a single sample of each control level was tested once every 24 hours. The SARS-CoV-2 IgG assay uses chemiluminescent microparticle immunoassay (CMIA) technology to detect IgG antibodies to SARS-CoV-2 in human serum and plasma. The test has a high sensitivity (100%) and specificity (over 99.6%) for the detection of SARS-CoV-2 specific IgG antibodies 17 days after infection [9, 10].

### Laboratory data entry

Separate EpiCollect5 forms, in duplicate, were used for entering laboratory results data. The forms were independently filled by two trained personnel. Data from the two forms were checked for consistency by a third person and in case of any discrepancy, the source data was referred for any needed correction. The unique identification number was used to link the interview information and laboratory results data.

### Definitions used

Participants were labeled as "IgG positive" if the index value for SARS-CoV-2 specific IgG was above 1.40, as suggested by the manufacturer [9]. Else, they were labeled "IgG negative".

### Statistical analysis

For the presentation of age data, age was grouped into 20-year age intervals. The occupation was categorized as 'at-risk occupation' and 'low-risk occupation' depending upon the perceived risk of exposure to a known or unknown SARS-COV-2 positive case. Categorical variables were summarized as frequency and percentage. Overall, unadjusted seroprevalence was reported as a percentage along with its 95% confidence interval. Given the nature of participant selection, we report age- and gender-standardized seroprevalence to minimize, if not nullify, the resultant bias. The Sample Registration System (SRS) Statistical Report 2018 [11] provides the latest available percentage distribution of the population of Jammu & Kashmir by age and gender. We used figures for urban areas in the report to calculate weights for reporting age- and gender-standardized seroprevalence. The details are in the supplementary material (S1 Table).

To identify potential factors associated with SARS-CoV-2 seropositivity, we used logistic regression analysis and reported unadjusted and age- and gender-adjusted odds ratio with a 95% confidence interval.

We estimated the number of infections till two weeks before the study period, i.e., 15th June 2020 to 30th June 2020, by applying the age- and gender-specific seroprevalence rates found in the study to the projected population of the District for the year 2020 using 2011 census data [12] and growth rates from the Sample Registration System (SRS) [13]. Details are in the supplementary material (S2 Table).

Stata version 15.1 (StataCorp. 2017. Stata Statistical Software: Release 15. College Station, TX: StataCorp LLC.) was used for data analysis.

### Results

Over the study period, 3031 eligible persons were invited for participation in the study. The refusal rate was 3.6% (108/3031). Two thousand nine hundred twenty-three blood samples were collected, but demographic data was missing for 17 participants. We analyzed data from 2906 participants (Fig 1).

Only 90/2906 (3.1%) of the participants were ≥70 years of age. There was almost equal representation from males and females. Among females, 475/1443 (32.9%) were pregnant at the time of the interview. 115/2906 (4.0%) reported ILI symptoms in the four weeks preceding

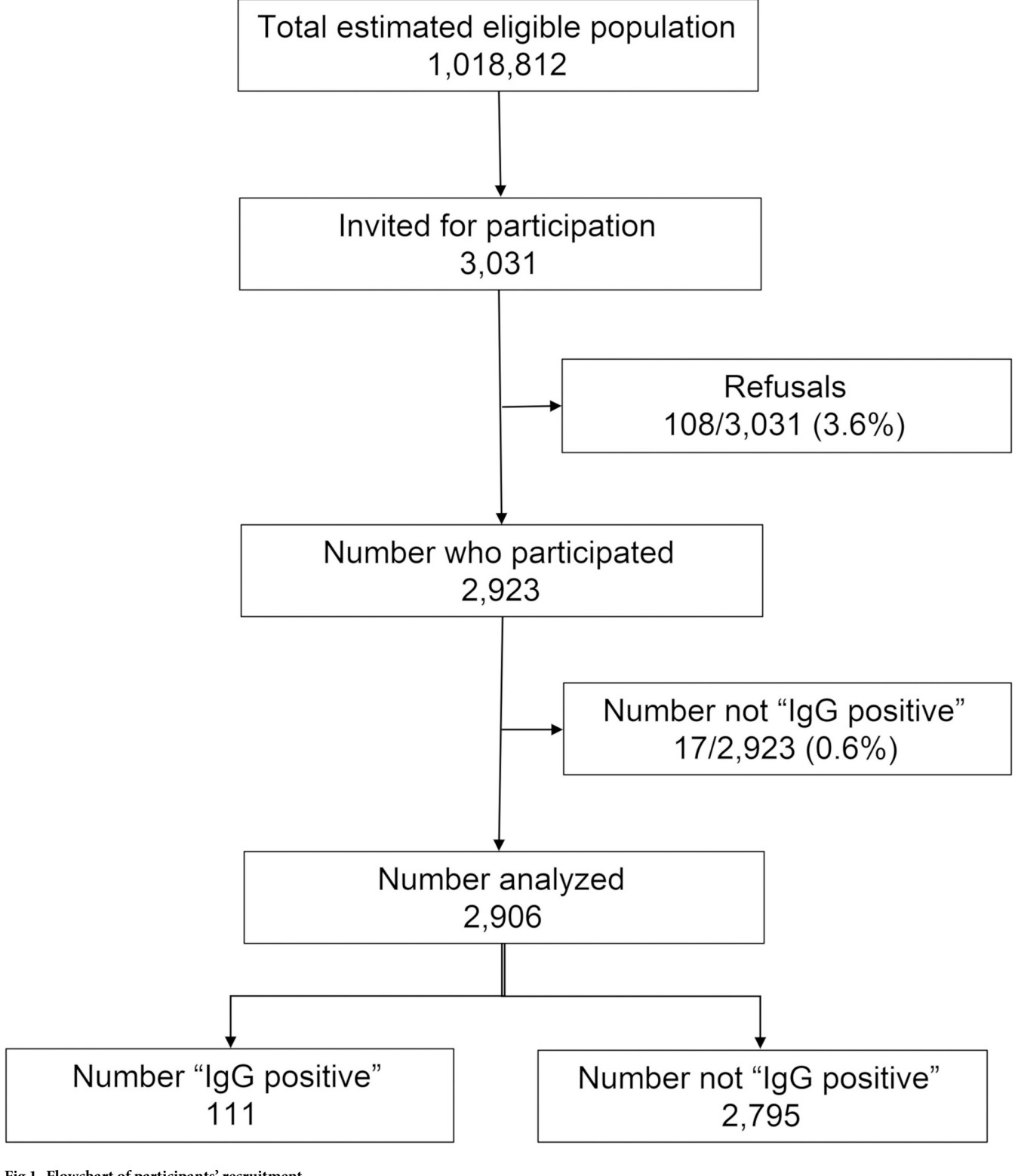

**Fig 1. Flowchart of participants' recruitment.**

the interview, 141/2906 (4.9%) reported having contact with a known SARS-CoV-2 positive case, 361/2906 (12.4%) had been tested for SARS-CoV-2 infection using RT-PCR, and 91/2906 (3.1%) had been placed under quarantine (Table 1).

**Table 1. Seroprevalence of SARS-CoV-specific IgG antibodies by participant characteristics.**

| Participant characteristics | Number of participants | Number of participants IgG Positive | Seroprevalence (95% CI) |
|---|---|---|---|
| Overall | 2906 | 111 | 3.8% (3.2–4.6) |
| | | | 3.6% (2.9–4.3) [a] |
| Age (years) | | | |
| <30 | 836 | 17 | 2.0% (1.3–3.2) |
| 30–49 | 1424 | 59 | 4.1% (3.2–5.3) |
| 50–69 | 556 | 32 | 5.8% (4.1–8.0) |
| ≥70 | 90 | 3 | 3.3% (1.1–9.8) |
| Gender | | | |
| Male | 1463 | 63 | 4.3% (3.4–5.5) |
| Female | 1443 | 48 | 3.3% (2.5–4.4) |
| Pregnant | | | |
| Yes | 475 | 11 | 2.3% (1.3–4.1) |
| No | 968 | 37 | 3.8% (2.8–5.2) |
| Occupation | | | |
| Stays home | 1434 | 54 | 3.8% (2.9–4.9) |
| Health-care provider | 355 | 17 | 4.8% (3.0–7.6) |
| Administrative | 186 | 7 | 3.8% (1.8–7.7) |
| Essential services | 179 | 9 | 5.0% (2.6–9.4) |
| Shopkeeper | 132 | 7 | 5.3% (2.5–10.7) |
| Laborer | 97 | 2 | 2.1% (0.5–7.9) |
| Media | 44 | 0 | - |
| Hotel staff | 37 | 4 | 10.8% (4.1–25.5) |
| Others | 442 | 11 | 2.5% (1.4–4.4) |
| ILI symptoms in the last four weeks | | | |
| Yes | 115 | 14 | 12.2% (7.3–19.5) |
| No | 2791 | 97 | 3.5% (2.9–4.2) |
| Any comorbidity | | | |
| Yes | 622 | 24 | 3.7% (2.5–5.5) |
| No | 2173 | 87 | 3.8% (3.1–4.7) |
| History of travel outside Kashmir since 1st January 2020 | | | |
| Yes | 104 | 4 | 3.8% (1.5–9.8) |
| No | 2802 | 107 | 3.8% (3.2–4.6) |
| History of contact with a known SARS-CoV-2 positive case | | | |
| Yes | 141 | 8 | 5.7% (2.9–10.9) |
| No | 2765 | 103 | 3.7% (3.1–4.5) |
| Ever tested for SARS-CoV-2 infection by RT-PCR | | | |
| Yes | 361 | 21 | 5.8% (3.8–8.8) |
| No | 2545 | 90 | 3.5% (2.9–4.3) |
| Ever put under quarantine | | | |
| Yes | 91 | 8 | 8.8% (4.5–16.6) |
| No | 2815 | 103 | 3.7% (3.0–4.4) |

[a] Age-and gender-standardized seroprevalence

Of 135 participants who reported symptoms compatible with COVID-19 since the four weeks preceding the interview (including any current symptoms), only 38 (28.1%) were tested by RT-PCR. Among asymptomatic, 97/2771 (3.5%) were IgG positive (Table 2).

Seventy-eight participants (78/2906 = 2.7%) visited the hospital for consultation for COVID-19-like symptoms (Table 3).

Hypertension (353/2906 = 12.1%) was the most common comorbidity followed by hypothyroidism (248/2906 = 8.5%), and diabetes mellitus (164/2906 = 5.6%) (Table 4).

The overall, unadjusted prevalence of IgG antibodies against SARS-CoV-2 (seroprevalence) in the study sample was 3.8% (95% CI 3.2%-4.6%) (Table 1). The age-and gender-adjusted seroprevalence was 3.6% (95% CI 2.9%-4.3%) (S3 Table).

Young people, those with a history of ILI symptoms in the four weeks preceding the interview, and those ever placed under quarantine were found to have significantly higher odds of the presence of SARS-CoV-2 specific IgG antibodies (Table 5).

During the two weeks before the start of the study (15th June 2020 to 30th June 2020), the estimated cumulative number of SARS-CoV-2 infections in the area was 36677 (95% CI 29546–43809) (S2 Table). When adjusted for sensitivity and specificity of the laboratory test kit [14], the number of infections comes down to 32602 (95% CI 25470–39734) (S2 Table). [Prevalence adjusted for sensitivity and specificity of the test was calculated as: (Age- and gender-adjusted prevalence + Test specificity– 1) ÷ (Test sensitivity + Test specificity– 1)].

The cumulative number of RT-PCR confirmed cases till 15th June 2020 was 538 in District Srinagar, and the number was 932 till 30th June 2020. The mid-interval (median) cumulative number of RT-PCR confirmed cases in the two weeks preceding the study was 703 (Fig 2). The infection-to-known-case ratio was thus between 32602/932 = 35 (95% CI 25470/932 = 27 to 39734/932 = 43) to 32602/538 = 61 (95% CI 25470/538 = 47 to 39734/538 = 74). The median estimate of the infection-to-known-case ratio was 32602/703 = 46 (95% CI 25470/703 = 36 to 39734/703 = 57).

## Discussion

The results of our study provide a rough estimate of the prevalence of IgG antibodies against SARS-CoV-2 in District Srinagar. Nearly 3.6% of the population showed evidence of recent SARS-CoV-2 infection. A large proportion of the population is, thus, still susceptible to the infection. Approximately three months after reporting the first COVID-19 case, the epidemic appears to be in the initial stages, and more cases of COVID-19 are expected. The number of confirmed cases of SARS-CoV-2 infection has been on the increase ever since we completed our study (Fig 2).

A large proportion of infections remain unknown. For every RT-PCR confirmed case, there are about 46 infections in the population. This figure is higher than that reported from Switzerland [15]. Interestingly, only 28% of participants in our study with COVID-

**Table 2. Status of RT-PCR testing and seropositivity by COVID-19 compatible symptoms.**

| | | Ever symptomatic [a] | Never symptomatic [a] |
|---|---|---|---|
| | | (n = 135) | (n = 2771) |
| Tested by RT-PCR | Yes | 38 (28.1%) | 323 (11.7%) |
| | No | 97 (71.9%) | 2448 (88.3%) |
| SARS-CoV-2 specific IgG | Positive | 14 (10.4%) | 97 (3.5%) |
| | Negative | 121 (89.6%) | 2674 (96.5%) |

[a] Refers to symptoms compatible with COVID-19 during the four weeks preceding the study including any current symptoms

**Table 3.  COVID-19 like symptoms reported by study participants at the time of hospital visit (n = 78).**

| COVID-19 like symptom | Number of participants |
|---|---:|
| Fever | 54 |
| Cough | 41 |
| Sore throat | 27 |
| Body aches | 26 |
| Shortness of breath | 12 |
| Running nose | 9 |
| Headache | 5 |
| Anosmia | 2 |
| Diarrhea | 1 |

19 like symptoms were tested by RT-PCR (Table 2). Robust mechanisms for testing of COVID-19 suspects need to be developed and implemented to decrease the number of unknown infections. Of the 111 participants who tested positive for SARS-CoV-2 specific IgG, 97 did not report any history of COVID-19 like symptoms. This finding is compatible with what we know about SARS-CoV-2 infections–a majority of them remain asymptomatic [6, 7].

Several seroprevalence studies conducted across the world have reported prevalence ranging from 1% in California to 23% in Delhi [15–22]. The prevalence estimates depend on the stage of the epidemic in the area at the time of the study and the accuracy of the antibody detection test used.

We used CMIA to detect SARS-CoV-2 specific IgG antibodies. CMIA-based IgG tests have high sensitivity and specificity for the detection of SARS-CoV-2 infection when the samples have been taken two weeks after the onset of symptoms [9, 23–28]. There is, however, a possibility of cross-reaction with human endemic coronaviruses, which can lead to false-positive results making a diagnosis of SARS-CoV-2 infection by antibody detection less specific [9, 23, 29].

IgG antibodies appear, on an average, 10–11 days after symptoms [15] or two weeks after infection and are maintained at a high level for an extended period [24, 25]. Immediately after infection, the IgG titers are negative and thus do not help in the diagnosis of the infection in the early stage. A combination of IgM and IgG antibody tests is more helpful [27]. Detection of infection by the use of SARS-CoV-2 specific IgM and IgG antibodies has several advantages. As compared to RT-PCR based detection of infection, the antibody-based tests are cheaper and faster. They also pose less danger of infection for health workers since patients may disperse the virus during respiratory sampling. Also, blood samples show reduced heterogeneity compared to respiratory specimens [23]. Besides, the presence of IgG antibodies gives a clue to the presence of humoral immunity to SARS-CoV-2. However, both B and T cells may provide immune-mediated protection to viral infection [26].

We did not find any significant difference in seroprevalence among males and females (OR 1.2, 95% CI 0.8–1.8) (Table 4). Similar findings have been reported elsewhere in the literature [17, 30]. People 30–69 years of age had a 2–3 times higher odds of the presence of SARS-CoV-2 specific IgG antibodies as compared to the young (<30 years) (Table 4). The lower seroprevalence in the old (≥70 years) points to the success of social distancing measures adopted by families, especially in the context of the elderly population, something the local health authorities have been stressing upon given the higher chances of more severe disease among the elderly. However, the possibility of immunosenescence among older adults cannot be ruled out and needs further investigation [31].

**Table 4. Comorbidities in the study participants.**

| Comorbidity | Number of participants | Percentage |
|---|---|---|
| Hypertension | 353 | 12.1% |
| Hypothyroidism | 248 | 8.5% |
| Diabetes mellitus | 164 | 5.6% |
| Chronic obstructive lung disease | 15 | 0.5% |
| Coronary heart disease | 14 | 0.5% |
| Chronic kidney disease | 9 | 0.3% |
| Cerebrovascular disease | 7 | 0.2% |
| Cancer | 6 | 0.2% |
| Chronic liver disease | 3 | 0.1% |
| Other comorbidities | 53 | 1.8% |
| No comorbidities | 2260 | 77.8% |

The presence of ILI symptoms in the recent past was the factor most strongly associated with the presence of SARS-CoV-2 specific IgG antibodies. People with a recent history of ILI symptoms had a 3.7 times higher chance of showing evidence of SARS-CoV-2 infection as compared to those without such history. Another factor significantly related to evidence of SARS-CoV-2 infection was placement under quarantine. During the first few months of the epidemic in the District, the authorities placed all SARS-CoV-2 suspects and their close contacts under 'administrative quarantine' at a designated place. There is a possibility that infection in such cases could be because of their contact with the primary suspect or contact during their stay at the quarantine facility. Nevertheless, having been under quarantine almost doubled the odds of SARS-CoV-2 infection as compared to those who did not need such quarantine.

## Limitations

The selection of study participants was not completely random, and this could have led to an overestimation of the seroprevalence estimates. A random population-based selection of participants was not feasible owing to the lockdown and human resource constraints in an already over-burdened healthcare system. We, hopefully, reduced the bias by providing age- and gender-standardized estimates, but could not nullify or estimate the bias. The low prevalence of SARS-CoV-2 in our setting and the possibility of false-positive results may inflate our

**Table 5. Unadjusted and age- and gender-adjusted odds ratio of seropositivity by participant characteristics.**

| | Unadjusted | | Age- and gender-adjusted | |
|---|---|---|---|---|
| | Odds ratio (95% CI) | p-value | Odds ratio (95% CI) | p-value |
| Age (years) | | | | |
| <30 | 1 (Reference) | - | 1 (Reference) | - |
| 30–49 | 2.1 (1.2–3.6) | 0.009 | 2.0 (1.2–3.5) | 0.011 |
| 50–69 | 2.9 (1.6–5.4) | <0.001 | 2.8 (1.6–5.2) | 0.001 |
| ≥70 | 1.7 (0.5–5.8) | 0.425 | 1.6 (0.5–5.6) | 0.464 |
| Male | 1.3 (0.9–1.9) | 0.169 | 1.2 (0.8–1.8) | 0.495 |
| At-risk occupation | 1.6 (1.1–2.4) | 0.023 | 1.5 (1.0–2.3) | 0.072 |
| ILI symptoms | 3.8 (2.1–7.0) | <0.001 | 3.7 (2.0–6.7) | <0.001 |
| History of contact with a known SARS-CoV-2 positive case | 1.6 (0.7–3.3) | 0.243 | 1.6 (0.7–3.3) | 0.244 |
| Ever put under quarantine | 2.5 (1.2–5.4) | 0.015 | 2.7 (1.3–5.8) | 0.011 |

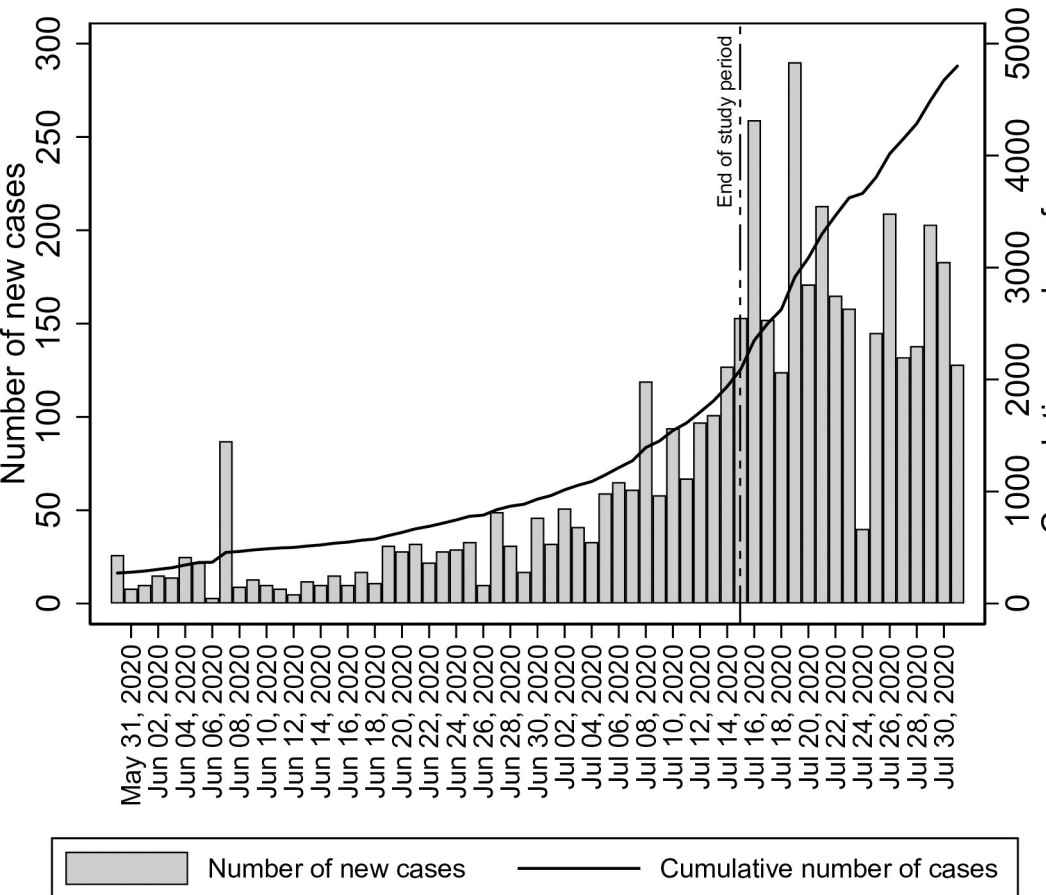

**Fig 2. Daily new cases of SARS-CoV-2 and the cumulative number of cases in District Srinagar, June-July 2020.**

prevalence estimates. We provide test-performance adjusted estimates of the number of infections in addition to age- and gender- standardization to reduce such bias. Detection of SARS-CoV-2 specific IgM antibodies and simultaneous RT-PCR could have provided a better estimate of the current infection rate.

## Conclusions

Our study provides estimates of SARS-CoV-2 infection in District Srinagar. The findings of our study suggest that there are many people with unknown infection in the community. For every known case, there are approximately 46 unknown infections. A sizeable number of symptomatic individuals do not receive RT-PCR testing for early diagnosis. The case-detection and contact-testing exercise should be intensified to allow the detection of unknown infections in the community. The study findings further suggest that a large proportion of the population is still susceptible to SARS-CoV-2 infection, and the number of cases may increase. There is a need to ensure the availability of treatment facilities in hospitals across the District. Infection surveillance measures need to be developed to devise informed public health measures.

## Supporting information

**S1 Data. List of hospitals across District Srinagar.**
(XLSX)

**S2 Data.**
(DTA)

**S1 Table. Age and gender distribution of the population in District Srinagar and the estimated population in 2020.**
(XLSX)

**S2 Table. Estimated number of infections in District Srinagar.**
(XLSX)

**S3 Table. Age- and gender-standardized seroprevalence.**
(LOG)

## Acknowledgments

We are grateful to Prof. Samia Rashid, Principal, Government Medical College, Srinagar; Shahid Choudhary, District Commissioner, Srinagar; the Department of Biochemistry, Government Medical College, Srinagar; Directorate of Health Services, Kashmir; Chief Medical Officer, Srinagar; all Block Medical Officers and Zonal Medical Officers of District Srinagar; all healthcare workers, and interns involved in this study. We thank the study participants for understanding the importance of this study and giving their time and consent for participation.

## Author Contributions

**Conceptualization:** S. Muhammad Salim Khan, Mariya Amin Qurieshi, Inaamul Haq.

**Data curation:** Mariya Amin Qurieshi, Inaamul Haq, Arif Akbar Bhat, Sahila Nabi, Nisar Ahmad Ganai, Nazia Zahoor, Auqfeen Nisar, Iqra Nisar Chowdri, Tanzeela Bashir Qazi, Rafiya Kousar, Abdul Aziz Lone, Iram Sabah, Shahroz Nabi, Ishtiyaq Ahmad Sumji, Misbah Ferooz Kawoosa, Shifana Ayoub.

**Formal analysis:** Mariya Amin Qurieshi, Inaamul Haq.

**Funding acquisition:** S. Muhammad Salim Khan.

**Investigation:** Mariya Amin Qurieshi, Inaamul Haq, Sabhiya Majid.

**Methodology:** S. Muhammad Salim Khan, Mariya Amin Qurieshi, Inaamul Haq, Sabhiya Majid.

**Project administration:** S. Muhammad Salim Khan, Mariya Amin Qurieshi, Inaamul Haq, Sabhiya Majid.

**Resources:** Sabhiya Majid, Arif Akbar Bhat, Sahila Nabi, Nisar Ahmad Ganai, Nazia Zahoor, Auqfeen Nisar, Iqra Nisar Chowdri, Tanzeela Bashir Qazi, Rafiya Kousar, Abdul Aziz Lone, Iram Sabah, Shahroz Nabi, Ishtiyaq Ahmad Sumji, Misbah Ferooz Kawoosa, Shifana Ayoub.

**Software:** Mariya Amin Qurieshi, Inaamul Haq.

**Supervision:** S. Muhammad Salim Khan, Mariya Amin Qurieshi, Inaamul Haq, Sabhiya Majid, Arif Akbar Bhat, Sahila Nabi, Nisar Ahmad Ganai, Nazia Zahoor, Auqfeen Nisar, Iqra Nisar Chowdri, Tanzeela Bashir Qazi, Rafiya Kousar, Abdul Aziz Lone, Iram Sabah, Shahroz Nabi, Ishtiyaq Ahmad Sumji, Misbah Ferooz Kawoosa, Shifana Ayoub.

**Validation:** Mariya Amin Qurieshi, Sabhiya Majid, Arif Akbar Bhat.

**Writing – original draft:** Mariya Amin Qurieshi, Inaamul Haq.

**Writing – review & editing:** S. Muhammad Salim Khan, Mariya Amin Qurieshi, Inaamul Haq, Sabhiya Majid, Arif Akbar Bhat, Sahila Nabi, Nisar Ahmad Ganai, Nazia Zahoor, Auqfeen Nisar, Iqra Nisar Chowdri, Tanzeela Bashir Qazi, Rafiya Kousar, Abdul Aziz Lone, Iram Sabah, Shahroz Nabi, Ishtiyaq Ahmad Sumji, Misbah Ferooz Kawoosa, Shifana Ayoub.

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
