## [Decision Letter · Decision Letter 0]

13 Oct 2020

PONE-D-20-27283

Seroprevalence of SARS-CoV-2 specific IgG antibodies in District Srinagar, northern India – a cross-sectional study

PLOS ONE

Dear Dr. Inaamul Haq,

Thank you for submitting your manuscript to PLOS ONE. After careful consideration, we feel that it has merit but does not fully meet PLOS ONE’s publication criteria as it currently stands. Therefore, we invite you to submit a revised version of the manuscript that addresses the points raised during the review process.

We look forward to receiving your revised manuscript.

Kind regards,

Daniela Flavia Hozbor

Academic Editor

PLOS ONE

Journal Requirements:

Reviewers' comments:

Reviewer's Responses to Questions

**Comments to the Author**

1. Is the manuscript technically sound, and do the data support the conclusions?

Reviewer #1: Yes

Reviewer #2: Yes

2. Has the statistical analysis been performed appropriately and rigorously? 

Reviewer #1: Yes

Reviewer #2: Yes

3. Have the authors made all data underlying the findings in their manuscript fully available?

Reviewer #1: Yes

Reviewer #2: Yes

4. Is the manuscript presented in an intelligible fashion and written in standard English?

Reviewer #1: Yes

Reviewer #2: Yes

5. Review Comments to the Author

Reviewer #1: Congrats on an informative study.

I am not familiar with "cold chain" and you may want to define "test med", procedure.

What was the performed RT-PCR sensitivity/specificity?

Knowing the time between the onset of symptoms and your serology test would be helpful in light of the lack or post-infection immunity for discussion and future reference/research.

Good idea on bringing the IgM issues/time line and R&D recommendation to Pharma.

Reviewer #2: Very nice article. Well written. Would like to know how they did the adjustment for specificity of the test in getting their overall positivity rate rather than just seeing the reference. Also, would like to see how the test performed in individuals who had been RT-PCR positive. I can’t find that in the primary paper. Also, I would prefer to see the methods you used to adjust for test specificity rather than just seeing the reference.

6. PLOS authors have the option to publish the peer review history of their article (what does this mean?). If published, this will include your full peer review and any attached files.

Reviewer #1: No

Reviewer #2: No

---

## [Author Response · Author response to Decision Letter 0]

21 Oct 2020

The response to reviewer comments has been uploaded as "Response to Reviewers" file. All reviewer comments have been responded.

---

## [Editor Report · Decision Letter 1]

26 Oct 2020

Seroprevalence of SARS-CoV-2 specific IgG antibodies in District Srinagar, northern India – a cross-sectional study

PONE-D-20-27283R1

Dear Dr. Inaamul Haq,

We’re pleased to inform you that your manuscript has been judged scientifically suitable for publication and will be formally accepted for publication once it meets all outstanding technical requirements.

Kind regards,

Daniela Flavia Hozbor

Academic Editor

PLOS ONE
---

## [Editor Report · Acceptance letter]

3 Nov 2020

PONE-D-20-27283R1 

Seroprevalence of SARS-CoV-2 specific IgG antibodies in District Srinagar, northern India – a cross-sectional study 

Dear Dr. Haq:

I'm pleased to inform you that your manuscript has been deemed suitable for publication in PLOS ONE. Congratulations! Your manuscript is now with our production department. 

Kind regards, 

on behalf of

Dr. Daniela Flavia Hozbor 

Academic Editor

PLOS ONE